# Ti and its alloys as examples of cryogenic focused ion beam milling of environmentally-sensitive materials

Yanhong Chang[1], Wenjun Lu[1], Julien Guénolé [2], Leigh T. Stephenson[1], Agnieszka Szczpaniak[1], Paraskevas Kontis [1], Abigail K. Ackerman[3], Felicity F. Dear[3], Isabelle Mouton[1], Xiankang Zhong [1], Siyuan Zhang[1], David Dye [3], Christian H. Liebscher[1], Dirk Ponge[1], Sandra Korte-Kerzel [2], Dierk Raabe[1] & Baptiste Gault [1,3]

Hydrogen pick-up leading to hydride formation is often observed in commercially pure Ti (CP-Ti) and Ti-based alloys prepared for microscopic observation by conventional methods, such as electro-polishing and room temperature focused ion beam (FIB) milling. Here, we demonstrate that cryogenic FIB milling can effectively prevent undesired hydrogen pick-up. Specimens of CP-Ti and a Ti dual-phase alloy (Ti-6Al-2Sn-4Zr-6Mo, Ti6246, in wt.%) were prepared using a xenon-plasma FIB microscope equipped with a cryogenic stage reaching −135 °C. Transmission electron microscopy (TEM), selected area electron diffraction, and scanning TEM indicated no hydride formation in cryo-milled CP-Ti lamellae. Atom probe tomography further demonstrated that cryo-FIB significantly reduces hydrogen levels within the Ti6246 matrix compared with conventional methods. Supported by molecular dynamics simulations, we show that significantly lowering the thermal activation for H diffusion inhibits undesired environmental hydrogen pick-up during preparation and prevents pre-charged hydrogen from diffusing out of the sample, allowing for hydrogen embrittlement mechanisms of Ti-based alloys to be investigated at the nanoscale.

[1] Max Planck Institute für Eisenforschung GmbH, Max-Planck-Straße 1, 40237 Düsseldorf, Germany. [2] Institute of Physical Metallurgy and Metal Physics, RWTH Aachen University, Kopernikusstraße 14, 52074 Aachen, Germany. [3] Department of Materials, Royal School of Mines, Imperial College, Prince Consort Road, London SW7 2BP, UK. Correspondence and requests for materials should be addressed to Y.C. (email: y.chang@mpie.de) or to B.G. (email: b.gault@mpie.de)

Titanium (Ti) is widely used in the aerospace and chemical industries owing to its excellent fatigue-allowable strength-to-weight ratio and good corrosion resistance[1]. However, it has a high affinity for hydrogen, and hydrogen pick-up is often associated with reductions in toughness and ductility[2,3], leading to a risk of premature failure in safety-critical, high capital value systems. Owing to the very low solubility of hydrogen in the hexagonal close-packed (hcp) α-Ti phase, many Ti alloys are prone to hydride formation[4]. This explains its potential for hydrogen storage[5], but the precipitation of brittle hydrides at the nanoscale would also be expected to reduce toughness. Investigation of the behaviour of hydrogen within Ti alloys is hence of interest to further understand both hydrogen embrittlement[2–4] and hydrogen storage[5]. However, studying the mechanisms of hydride formation in Ti alloys is challenging because of the undesired hydrogen introduction during the preparation of specimens for microscopic observations. For instance, hydrides observable by optical microscopy have been reported to form during mechanical grinding and polishing in acidic conditions[6]. Preparation of thin foils for transmission electron microscopy (TEM) by electrochemical polishing, even at low temperatures in the range of −40 to −50 °C, has long been known to pose a risk of the introduction of hydrogen-related phases or hydrides along the interface between the α and β phases in Ti alloys[7,8]. On occasion, the face-centred cubic δ-hydride phase is even confused with being a new, face-centred cubic phase of titanium[9,10]. Broad ion milling is frequently used to assist in hydride-free TEM foil preparation. However, according to the work of Carpenter et al.[11], hydrogen uptake and hydride formation were also induced by 5 kV argon ion beam sputtering on Ti-thin foils.

Over the past decade, the use of focussed ion beam (FIB) techniques has become widespread for preparing specimens for microscopic investigations, including thin lamellae for TEM[12] and needle-shaped specimens for atom probe tomography (APT)[13]. The approach can be used to target specific features such as grain boundaries and interfaces with a spatial resolution of better than 100 nm[14]. However, FIB milling introduces structural defects because of beam-induced damage including the formation of an amorphous surface layer, point defects (e.g., vacancies) and extended defects (e.g., dislocation lines and loops)[15,16]. Often, ion beam damage is simulated using the 'stopping range of ions in matter' (SRIM) package[17], by which the stopping distance and the resulting distribution of implanted incident ions and damage within a substrate can be evaluated. However, the model in this widely used package does not account for the crystalline nature of the target material, i.e., the material is assumed to be amorphous. However, the crystal structure of the specimen leads to, for example, channelling that strongly affects the final distribution of ions, and in particular the depth of implantation[18]. SRIM also ignores effects such as re-arrangement of the atoms locally that can cause recrystallization, healing the structure locally and thereby reducing or removing the amorphous layer that is typically formed, and leaving only structural defects such as vacancies and stacking faults. These effects are particularly relevant in metals[19], and the presence of vacancies and extended defects is likely to accelerate diffusional processes taking place subsequent to implantation. In contrast, molecular dynamics (MD) simulations have been shown to better reproduce experimental observations of the damage induced by the implantation of energetic ions into crystalline materials[20–22], and has been used to investigate FIB damage in detail[18].

Besides these common artefacts caused by FIB milling, it has been also reported that specimen preparation performed at room temperature has a high tendency to generate undesired hydrides during the preparation of TEM lamellae in both Ti and Zr alloys. Ding and Jones[23] demonstrated that hydrides form in the TEM lamellae of CP-Ti prepared by FIB milling, especially for the specimens cut from the water/acid solution polished bulk samples. Similar hydride formation in a Zr alloy during FIB thinning of TEM lamellae was also directly observed by Shen et al.[24]. In a recent study, we performed TEM analysis for structural investigation and APT for precise compositional analysis of CP-Ti and a set of Ti-based alloys[25]. Our results showed again that significant hydrogen ingress and hydride formation was most likely induced during FIB-based preparation of the specimens at room temperature[25]. Undesired hydrogen introduction during sample preparation makes precise characterization of these alloys particularly challenging, specifically hindering the investigation of the hydrogenation and hydrogen embrittlement behaviour of entire classes of alloys, despite the possibility of dedicated hydrogen or deuterium charging[26–29].

Here, we demonstrate how FIB conducted at temperature below 150 K, i.e., cryogenic focussed ion beam (cryo-FIB), allows to prevent undesired hydrogen pick-up and hydride formation in Ti alloys during sample preparation. Cryo-FIB milling and specimen handling has been extensively used in the biological sciences for electron microscopy investigations[30], and is increasingly applied for atom probe investigations[31,32]. We first carefully characterize hydrogen pick-up and hydride formation in CP-Ti and alloyed-Ti induced by both electro-polishing and room temperature FIB with Ga and Xe plasma sources (conventional Ga-FIB/Xe-PFIB (Ga focussed ion beam/Xe plasma focussed ion beam)). We then show that cryogenic Xe plasma FIB (cryo-PFIB) can help avoid the introduction of artefactual hydrides. We expect that the improvement arises from changes in the ion implantation and sputtering processes. MD simulations were hence performed to investigate the implantation of Ga and Xe ions in pure Ti, and associated structural damage. The damage profiles are rather similar, and hydrogen possibly located at or near the surface will likely be implanted within the first few nanometres below the specimen's surface in both cases. However, the use of cryogenic temperature greatly limits or even suppresses thermally assisted hydrogen diffusional processes, which explains our observations. Cryogenic Ga-FIB is expected to offer the same benefit. Cryo-FIB thus appears as the only technique allowing for preparing suitable specimens with minimal hydrogen ingress.

## Results

**Microstructure characterization.** Figure 1a–d shows the TEM/scanning transmission electron microscopy (STEM) micrographs of grade 2 commercially pure CP-Ti prepared by electrochemical polishing, room temperature Ga-FIB milling, room temperature Xe plasma FIB and cryo-PFIB milling, respectively. As shown in Fig. 1a, in the electrochemically polished TEM specimen, 25 nm-thin plates of a secondary phase appear with a dark contrast in high-angle annular dark-field scanning transmission electron microscopy (HAADF-STEM). In this mode of operation, the contrast is directly related to the average atomic number (Z-contrast), which implies that this phase has an average lower mass number compared with the matrix. Further phase identification using selected area electron diffraction (SAED) confirms that the phase is a δ-hydride with a face-centred cubic lattice structure. The orientation relationship between the δ-hydride and the α-matrix is $\{110\}_{hydride}//\{01\bar{1}0\}_\alpha$; $<001>_{hydride}//<0001>_\alpha$, which is well known for the hydride phase induced by electrochemical polishing[8]. In specimens prepared at room temperature by means of Ga-FIB/Xe-PFIB, similar plate-shaped precipitates were observed, as readily visible in Fig. 1b, c. The 200–500 nm-thick δ-hydride platelets possess an orientation relationship $\{111\}_{hydride}//\{0001\}_\alpha$; $<110>_{hydride}//<11\bar{2}0>_\alpha$ with the host matrix, which has previously been reported for hydrides introduced during Ga-FIB

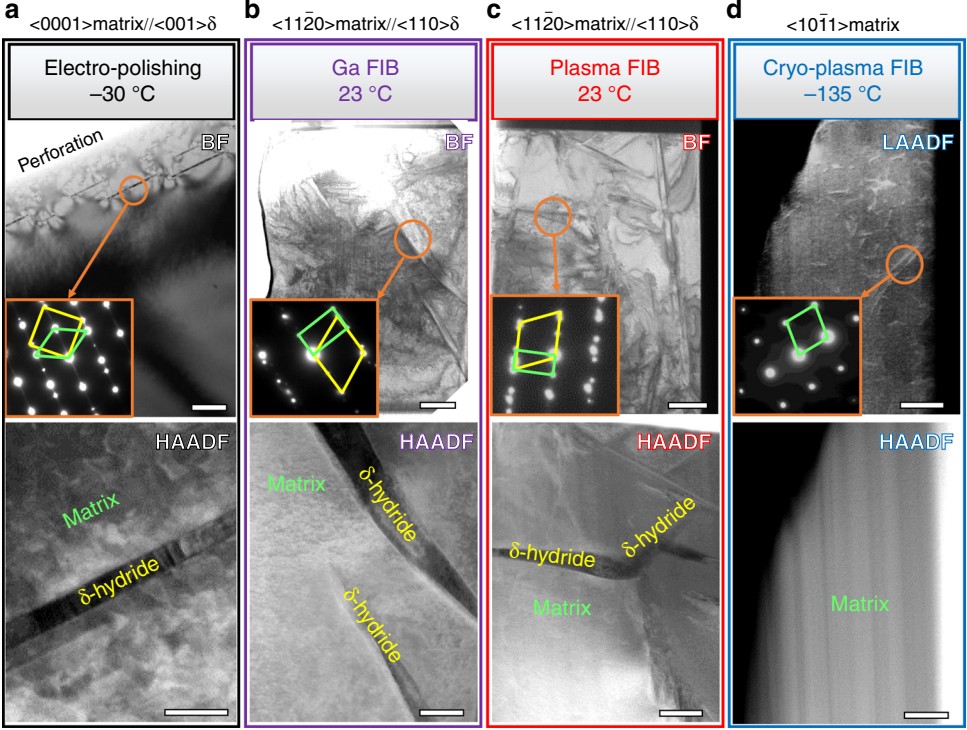

**Fig. 1** Transmission electron microscopy characterization. The bright-field TEM, SAED and STEM images for CP Ti prepared by **a** electro-polishing, **b** room temperature Ga-FIB, **c** room temperature Xe plasma FIB (Xe-PFIB) and **d** cryogenic Xe plasma FIB (cryo-PFIB). The scale bars represent 1 μm in the BF and 50 nm in the HAADF (**a**), 1 μm in the BF and 500 nm in the HAADF in (**b**, **c**), 500 nm in both the BF and the HAADF in (**d**). The inserts show the corresponding SAEDs indicating that the electron beam was close to the <0001> α, <11$\bar{2}$0> α, <11$\bar{2}$0> α and <10$\bar{1}$1> α zone axes for (**a**), (**b**), (**c**) and (**d**), respectively. Zone lines of each set of images are shown on top. The yellow and green lines in the insert SAEDs correspond to the diffraction reflections of δ-hydride and matrix, respectively. TEM transmission electron microscopy, SAED selected area electron diffraction, STEM scanning transmission electron microscopy, FIB focussed ion beam, HAADF high-angle annular dark field

milling at room temperature[23]. The HAADF micrographs obtained from specimens prepared by cryo-PFIB, however, contain only the Ti matrix (see Fig. 1d, the vertical lines relate to thickness variations and are an artefact of milling). The only microstructural feature that was revealed by strain contrast imaging in low-angle annular dark-field STEM (LAADF-STEM) mode were the dislocation lines, as shown in Fig. 1d, which were most likely present in the initial material due to processing. Our TEM observations confirm that hydride formation is induced during conventional Ga-FIB/Xe-PFIB milling at room temperature, which could be entirely inhibited by FIB milling at cryogenic temperature, −135 °C.

**Measuring H content by atom probe tomography**. APT measurements were performed to examine the hydrides and measure the level of solute hydrogen in the specimens prepared by either conventional Ga-FIB/Xe-PFIB milling or cryo-PFIB. Table 1 summarizes whether hydrides were detected, as well as the average amount of H (at.%) detected in the α-matrix and/or β phase of both CP-Ti and the commercial alloy Ti-6Al-2Sn-4Zr-6Mo (Ti6246, all in wt.%). A large number of specimens were prepared and measured, and hydrides with a composition of approximately 50 at.% H are found in approximately 90% of the specimens from CP-Ti prepared by conventional Ga-FIB/Xe-PFIB. In contrast, no hydride was ever encountered in CP-Ti specimens prepared by cryo-PFIB, consistent with the TEM observations. Four typical datasets of CP-Ti and Ti6246 alloy are shown in Fig. 2a, b for specimens prepared by conventional Ga-FIB and in Fig. 2c, d by cryo-PFIB. The corresponding mass spectrum for each dataset is also shown to reveal the number of detected ions, including

H-related species, $H^+$ at 1 Da and $H_2^+$ at 2 Da, as well as $Ti^{2+}$ ions at 23, 23.5, 24, 24.5 and 25 Da. For the specimens prepared by conventional FIB, the H-related peaks are higher than most $Ti^{2+}$ peaks, which represents the majority of detected ions. In contrast, for the specimens prepared by cryo-PFIB, the $H^+$ peak is small compared to the $Ti^{2+}$ peaks, and the $H_2^+$ peak is even smaller and sometimes invisible, as shown in the mass spectra of Fig. 2c, d. The detected hydrogen level in the α matrix of CP-Ti specimens, whether prepared by conventional FIB (Fig. 2a) or by cryo-PFIB (Fig. 2c), is below 2 at.%, which mostly originates from the residual gas in the ultra-high vacuum chamber of the APT microscope[25,33]. In the specimens of the two-phase Ti6246 alloy, prepared by conventional FIB milling, no hydride was detected, consistent with the observations of Ding and Jones[23] on dual-phase Ti-6Al-4V (Ti64, wt%) or β-phase Ti–5Mo–5V–3Cr (Ti-5553, wt%) TEM lamellae, and with our previous work[34–37]. However, a much higher amount of dissolved H was detected, 2–6 at.% H in the hexagonal α phase and 13–19 at.% H in the body-centred cubic β phase, consistent with our previous report[25]. In contrast, less than 2 at.% H is detected in both α phase and β phase of Ti6246 specimens prepared by cryo-PFIB, transferred either at ambient lab condition or under cryogenic ultra-high vacuum conditions, as shown in the H distribution map and composition profile in Fig. 2d. This value is again consistent with the residual level in the analysis chamber of APT. Our APT analyses of the H content in specimens prepared by cryo-PFIB and transferred via either route definitely prove that H pick-up in the β-containing Ti6246 alloy and hydride formation in CP-Ti are caused by the FIB milling at room temperature and these deleterious effects can be avoided by using cryo-PFIB preparation.

**Table 1 Comparison of H amount introduced by conventional Ga-FIB/Xe-PFIB and cryo-PFIB preparation**

| Specimen preparation methods | Hydride formation in CP-Ti | H content in α matrix in CP-Ti (at.%) | No. datasets obtained for CP-Ti {No. ion for each dataset (mill.)} | H content in α phase in Ti6246 (at.%) | H content in β phase in Ti6246 (at.%) | No. datasets obtained for Ti6246 {No. ion for each dataset (mill.)} |
|---|---|---|---|---|---|---|
| Conventional Ga-FIB/Xe-PFIB | Yes | ≤2 | 7 {3.6, 6.1, 37.7, 47.7, 32.7, 65.1, 80.3} | 2–6 | 13–19 | 8 {8.8, 7.1, 86.9, 16.4, 12.4, 30.9, 45.6, 55.2} |
| Cryo-PFIB | No | ≤2 | 7 {0.9, 1.1, 3.3, 13.2, 1.3, 5.3, 5.4} | ≤2 | ≤2 | 7 {12.8, 2.5, 20.2, 7.7, 34.7, 5.5, 5.2} |

We show atom probe tomography (APT) results of hydride visibility and the average H amount (at. %) detected in the matrix α and/or β phase of CP-Ti and Ti6246 alloy. All data obtained in high-voltage mode. The number of measured specimens and the number of ions (in millions) run in each specimen are provided
*Ga-FIB/Xe-PFIB* Ga focussed ion beam/Xe plasma focussed ion beam, *cryo-PFIB* cryogenic plasma focussed ion beam

**Molecular dynamics simulations**. During milling, the implantation of energetic ions into the specimen's structure induces surface damage, causes surface atoms to penetrate into the bulk and leads to the creation of crystalline defects. The presence of such radiation damage affects subsequent diffusion, for instance of H. We used MD in order to gain more detailed understanding of the effect of beam damage on the structural damage and the associated introduction of H into the material. Details of the implementation can be found in the Methods section. We report the final distribution of atoms after irradiation of a pure Ti surface and the peak temperature reached in the specimen locally, using a potential which was recently shown to yield accurate results in describing defects, plasticity and phase transitions in pure Ti[38]. To be compatible with times accessible by MD simulations, the time in between two incoming ions was only around 10 ps, which hence translates into a virtual current much higher than the effective currents applied experimentally. This limits the possible use of these results to gain insight into diffusion subsequent to implantation directly from this set of simulations.

Figure 3a is a series of MD snapshots at various doses during the process of implantation of Xe ions in (0001)-oriented pure Ti surface. The interaction of the incoming ions with those within the substrate leads to knock-on damage, causing the displacement of atoms from the substrate, and leading to the creation of vacancies and possibly self-interstitials, potentially leading to amorphisation of the surface[15]. In metals, where the defect mobility is high, these can annihilate or coalesce. Here, what is targeted was the observation of the intermixing of the surface layer with the substrate because of this knock-on damage. The atoms initially within 1 nm of the surface were coloured from yellow to purple according to their initial distance to the metal surface, whereas bulk atoms are shown in light purple. This allows direct estimation of the depth to which surface atoms may travel, as they are 'pushed' into the metal substrate by the incoming energetic ions. This is expected to mimic well the possible penetration of H-containing species adsorbed on the surface (see Methods).

Figure 3b is a plot of the initial vs. final distance to the surface for implantation, i.e., recoil implantation. If no atoms were displaced on average, all should lie on the solid black line. Simulations were performed for 30 kV singly charged Xe and Ga ions, both at normal and grazing incidence angle. The former corresponds to the case of the APT specimen preparation, the latter to the preparation of a TEM lamella. A slightly more pronounced displacement of the surface atoms is observed for the case of the Xe ions. The gap that appears at a small distance to the surface at grazing incidence angles is the result of the expected sputtering for such a configuration, although overall the sputtering remains negligible at such a relatively low dose (100 ions). Strong channelling at normal incidence is evidenced by the broad scatter of the recoil implantations around the

no-displacement line. Figure 3c shows a similar graph for simulations performed for singly charged Ga ions accelerated to 5 kV and 30 kV, both at normal incidence. With a lower incoming energy, the spread in the distribution of the surface atoms is reduced, as expected from previously published studies[39,40]. Only a low dose can be deposited during the short time of a simulation. It could hence be argued that the total energy deposited into the surface by the ionic irradiation in the MD simulations is too low to be representative. We hence decided to only use the higher acceleration voltage (30 kV) for our other simulations. Simulations investigating the effect of the base temperature of the substrate were performed with an acceleration voltage of 30 kV (the results shown in Fig. 3d). Decreasing the base temperature of the substrate leads to a slight reduction of the observed redistribution of the surface atoms for both Xe and Ga ions. Previous results from simulations of FIB milling reported extremely high peak temperatures localized where the incoming ions penetrate the metal[41], and our simulations exhibit similar local peak temperatures largely above the melting point but quenched back to room temperature within approximately 2 ps, and even faster at low temperature, resulting in a reduced level of damage.

**Sources of hydrogen**. The possible sources of hydrogen have been previously discussed[23–25]. First, Ti has a high affinity for hydrogen, and a large amount of hydrogen is expected to be adsorbed on the surface of bulk samples, which were subjected to mechanical grinding in water and polished using colloidal silica with $H_2O_2$. The work of Ding and Jones[23] reported that more hydrides formed in the conventional FIB-prepared TEM foil cut from the water/acid solution polished bulk sample than from a dry-ground sample. These results indicate that the presence of H before the final steps of thinning can affect the likelihood of preparation artefacts. Carpenter et al.[11] observed hydride formation in TEM foils of both CP-Ti and commercial Ti64 alloy induced by argon ion sputtering to perforation on a jet-electropolished 3-mm disc at ambient temperature with accelerating voltages of 5 keV. They suggested that one source of hydrogen could be the decomposition reaction of the hydrocarbon and of the moisture from the vacuum chamber stimulated by the energetic ion beam. During FIB preparation, there is an additional source of hydrogen. An organometallic precursor for Pt deposition, originating from the gas-injection system in the FIB, is used during the specimen preparation to protect the sample's surface or weld a pre-shaped specimen to a support. According to Wnuk et al.[42], electron beam irradiation may induce the decomposition of this Pt-containing organometallic precursor, accompanied by the evaporation of methane and dihydrogen gas. It is considered to be one of the major source for hydrogenation of Ti alloys[25] and Zr alloys[24] during room

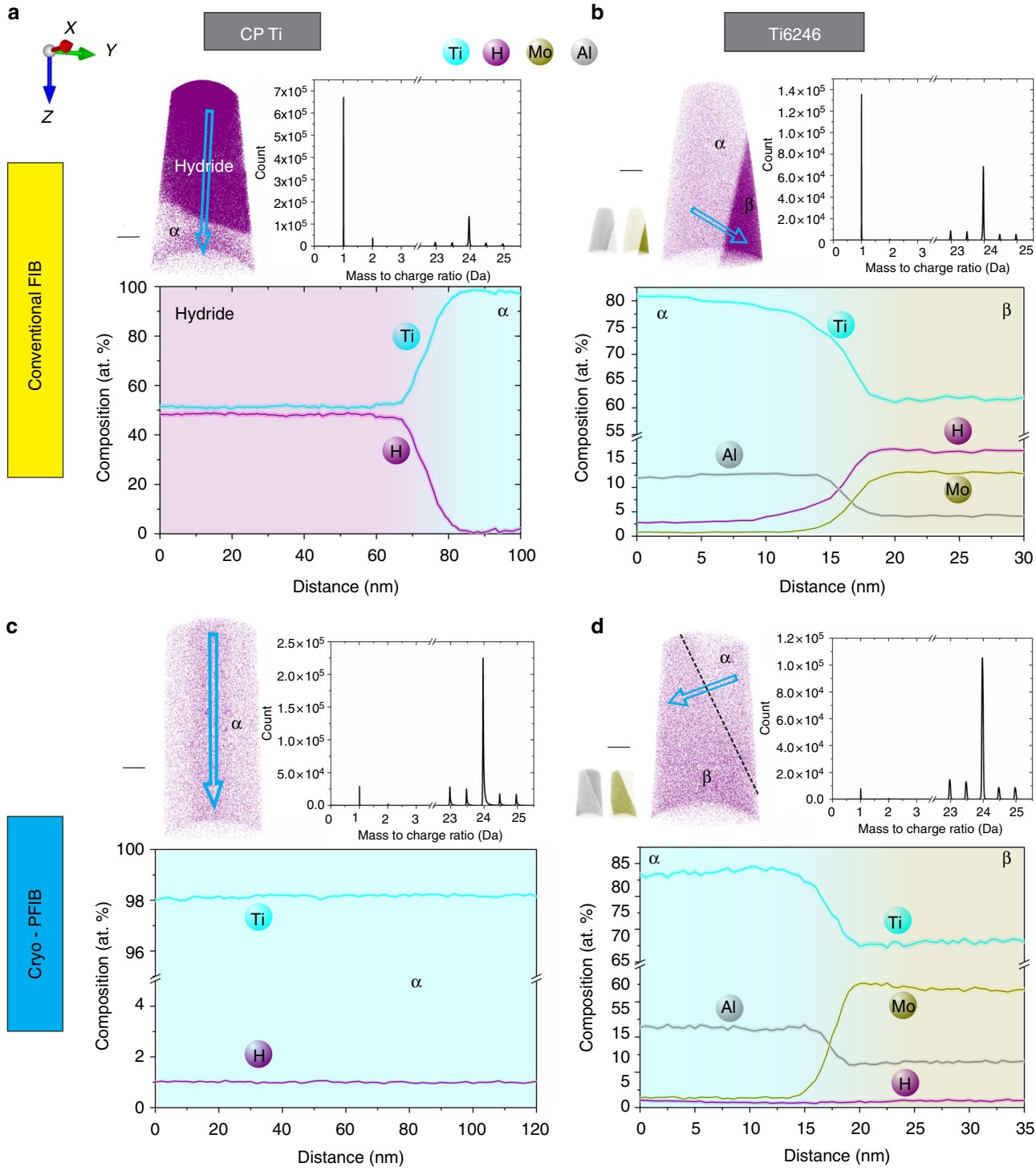

**Fig. 2** Atom probe microanalyses. Atom probe tomography (APT) H distribution map, mass spectrum and composition profile of **a** CP Ti and **b** Ti6246 specimens prepared by conventional focussed ion beam (FIB) at room temperature; **c** CP Ti and **d** Ti6246 specimens prepared by cryogenic plasma focussed ion beam (cryo-PFIB). The scale bars are all 20 nm

temperature Ga-FIB milling. Furthermore, surface sputtering during FIB milling constantly produces a highly reactive surface that may facilitate the adsorption of H onto the surface of the specimens, which can be introduced below the specimen's surface during implantation by the energetic ion beam. To identify the contribution of the residual gas from the high vacuum chamber of the PFIB, we tracked the chamber pressure during the cooling process, as shown in Fig. 4, and it only dropped from $3–6 \times 10^{-4}$ Pa at room temperature to $1–2 \times 10^{-4}$ Pa at

approximately $-135\,^{\circ}$C. Although the partial pressure of hydrogen is not specifically measured, it is unlikely that it would significantly fluctuate with such a minute decrease in the overall pressure and at a temperature above its sublimation temperature that is likely below 10 K at $10^{-4}$ Pa. However, at this temperature and pressure, water is expected to condensate on the colder parts and the partial pressure of water will therefore decrease[43], leading to a lower activity of the possible water splitting at the Ti surface.

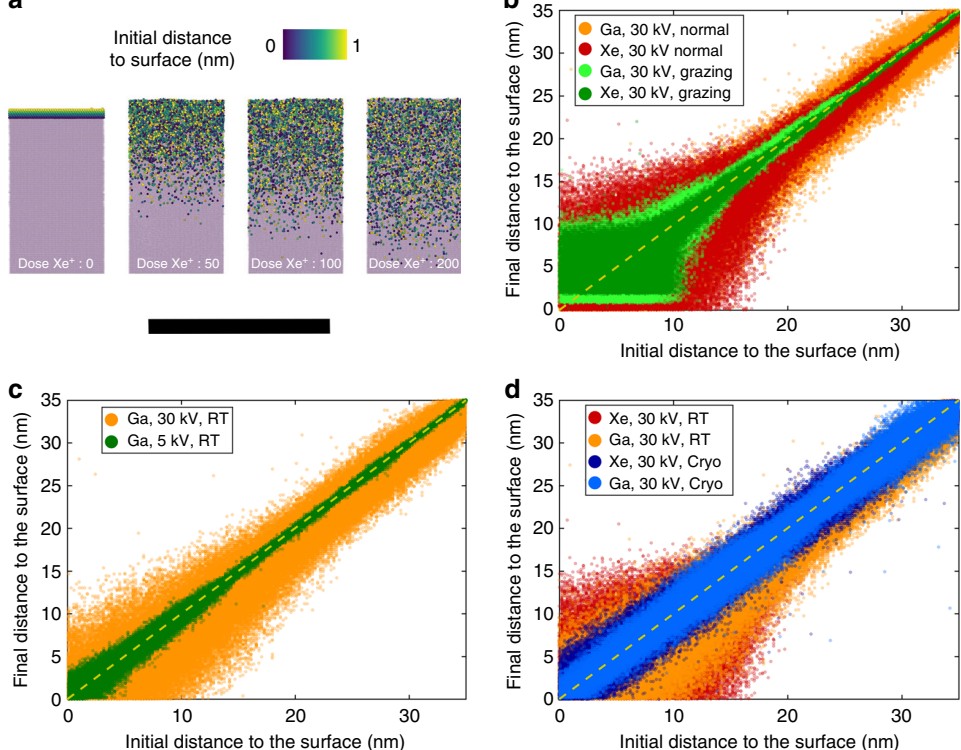

**Fig. 3** Molecular dynamics (MD) simulations. **a** MD snapshots at different stages of the irradiation process of Xe at 30 kV in (0001)-oriented pure Ti surface at normal incidence. Atoms initially within 1 nm of the surface are indicated by the colour scale. The bulk atoms are shown in light purple. The scale bar is 10 nm. **b** Position of Ti atoms after irradiation (final distance to surface) as a function of their position before irradiation (initial distance to surface). **c** Similar graph as in (**b**) following Ga implantation at acceleration voltages of 5 kV (pink) and 30 kV (blue). **d** Similar graph as in (**b**) following Xe and Ga implantation at room temperature (RT) (dark red and orange respectively) and at cryogenic temperature, i.e., 138 K (light and dark blue respectively)

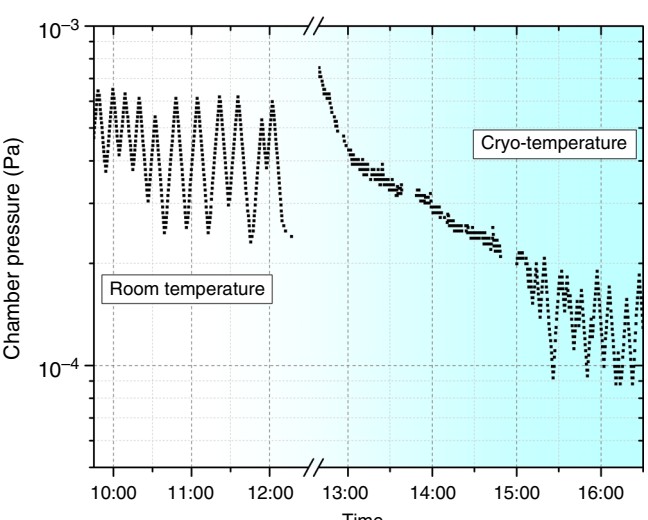

**Fig. 4** Pressure inside the scanning electron microscope. Profile of the pressure in our plasma focussed ion beam (PFIB) chamber from room temperature to cryogenic condition during the cooling process

**FIB-induced structural damage**. The milling process in the Xe-PFIB is slightly faster than by Ga-FIB for the materials used in our experiments. Consistently, our MD simulations with a grazing ion incidence angle produce higher sputtering rates with Xe than with Ga ions. Burnett et al.[44] also reported that the sputtering rate of Xe ion beam at 30 kV on Ti is slightly higher than that of Ga ion beam, as revealed via both experimental measurement and Monte

Carlo simulations using the SRIM package[17]. The MD simulations conducted here clearly show a small difference for atoms closer to the surface with Xe inducing slightly more damage, and hence possible higher introduction of H located at the specimen's surface. The maximum depth reached by atoms initially located at the surface, which are pushed inside by incoming Ga or Xe ions and the structural damage that they cause, is consistently below 15 nm at normal incidence, notwithstanding ion type and incidence angle. Ion channelling does not favour a deeper implantation of surface atoms. However, our TEM observations shown in Fig. 1 and APT measurements summarized in Table 1 show that both conventional Ga-FIB and Xe-PFIB milling at ambient temperature induce similar hydrogen pick-up and hydride formation. This is an indication that the ion species for FIB milling and the associated slight differences in sputtering rate and beam damage are not relevant for hydrogen ingress and hydride formation at room temperature in CP-Ti and Ti alloys.

At cryogenic temperature, −135 °C, the milling process and milling speed during specimen preparation showed no difference compared with Xe-PFIB milling at room temperature with the same FIB milling parameters, including ion beam energy, beam current and angle of incidence. Our MD simulations of ion implantation show a highly localized temperature increase, largely above the melting point, but that is quenched back to room temperature within approximately 2 ps. The beam damage on the surface is slightly reduced when decreasing the base temperature of the substrate. The diffusion of radiation-induced defects, e.g., vacancies and self-interstitials, after the initial beam damage is expected to be highly suppressed as well at cryogenic temperature. It is hence reasonable to conclude that FIB-induced structural damage is reduced at cryogenic temperature.

**Beam-induced heating**. Beam-induced heating was considered to play a critical role in accelerating hydrogen in-diffusion and enhancing hydride formation by both conventional broad ion beam (Ar ions) milling[11] and focused ion beam (Ga ions) milling[23]. In earlier work, it has been reported that conventional argon ion milling could heat TEM specimens up to around 400 °C[45] and further cause both microstructural and chemical instability in a wide range of materials including Au/Si specimens[46] and metallic alloys such as aluminium[47], copper and stainless steel[48]. Conducting liquid-N$_2$ (LN$_2$) cooling of the specimen stage during Ar$^+$ ion milling has proved to be an effective pathway to prevent artificial chemical instability by decreasing diffusivity in such cases. For example, the Au/Si interface migration caused by beam-induced heating could be eliminated by using a LN$_2$ cooled specimen stage[46]. Kenik[48] also showed that low temperature milling could effectively alleviate the loss of grain boundary segregating elements, such as bismuth in copper or phosphorous in stainless steel. With the same consideration, Carpenter et al.[11] suggested that cryo-cooling of specimen could be of significant benefit on preventing hydrogen pick-up and hydride formation during broad Ar ion milling of Ti and Zr materials.

However, experimentally, focussed ion beam milling is known to induce a much lower average temperature rise than broad ion milling. This is due to a reduction of 2–3 orders of magnitude in the incident beam power, according to the calculation put forward by Ishitani et al.[45,49]. The maximum local temperature rise, $\Delta T_{max}$, induced by focussed ion beams on a bulk specimen can be estimated by the following equation[49]:

$$\Delta T_{max} = V\,I/d\,k\,\pi^{0.5}, \tag{1}$$

where $V$ is the accelerating voltage of the ion beam, $I$ the beam current, $d$ the beam diameter and $k$ the thermal conductivity of the specimen ($k = 22\ \mathrm{W\,m^{-1}\,K^{-1}}$ for pure Ti). In our experiments, the lift-out procedures from bulk samples were all conducted with a beam current of 6–9 nA at an accelerating voltage of 30 kV at ambient temperature. From the above equation, the predicted maximum local heating during lift-out is less than 10 K for both conventional Ga-FIB and Xe plasma FIB. In our protocol, the lift-out is performed at room temperature followed by thinning or annular milling and sharpening at cryo-temperature. Consistent with this estimate, our observations suggest that the lift-out process is not relevant to the hydrogen pick-up.

During thinning or sharpening, the specimen is milled from 2–3 μm to below 100 nm in thickness for TEM lamellae and 50–100 nm in diameter for APT needles. According to the calculation of Ishitani and Kaga[49], the local heating of a sheet- or pillar-shaped specimen could be typically 2–4 times higher for a sheet and 10 or more times higher for the pillars compared to a bulk sample under the same conditions of beam current and accelerating voltage. The local temperature rise is expected to gradually increase with the decreasing size of the TEM or APT specimens during milling. Heating is likely mitigated by the progressive decrease in beam current from 0.46 nA to 24 pA during successive preparation steps. These considerations are relative to an increase of the average temperature of the specimen during milling, not to the very local peak temperature that is reached at or near an implanted ion, as seen above in the MD calculations.

**H ingress**. Hydrogen diffusivity in Ti critically depends on the sample's temperature[50]. At room temperature, hydrogen can diffuse over 300 nm within 5 min in α-Ti, and within seconds in β-Ti. Our MD simulations indicate that the hydrogen observed by

APT after room temperature preparation cannot originate from implantation alone, but must have diffused in through the structure. Hydrogen introduction is expected to be slightly facilitated by the high density of structural defects below the sputtered surface[15,16]. At room temperature, radiation-induced point defects, in particular vacancies, can diffuse and annihilate at the surface of the specimen, which could help introduce H[51]. However, since H diffuses mostly via an interstitial mechanism, the influence of crystalline defects on the long-range H diffusion is expected to be low at room temperature[51]. The diffusivity of H might be slightly enhanced by beam-induced heating, which is expected to be limited by the low currents and rather low dwell time used during milling. Therefore, it is the intrinsic fast diffusion of H that plays the critical role in the H ingress during conventional FIB milling. This also explains why hydrides are often observed at interfaces or grain boundaries that can act as fast diffusion paths for solutes.

At cryogenic temperature, beam damage and possible H implantation is confined within 5 nm on the surface, indicated by the MD simulation, and the mobility of the radiation-induced defects is expected to be constrained. When simply extrapolating the Arrhenius law to lower temperatures, hydrogen diffusivity in the α and β phases of Ti alloys at −135 °C would be 8–12 orders of magnitude slower than at room temperature. It would take approximately 300 h to diffuse over 10 nm into β-Ti, while years into α-Ti. There is a lack of available experimental data for the H diffusivity at cryogenic temperatures, but it is reasonable to assume the hydrogen diffusion in both α and β phases at cryogenic temperature is orders of magnitude lower than at room temperature. Cryo-cooling of the stage down to −135 °C substantially inhibits undesired in-diffusion of hydrogen from the environment in both α- and β-Ti, and hence prevents hydride formation in α-Ti and hydrogen accumulation in β-Ti during FIB milling. The processes of H pick-up during sharpening of an APT specimen by conventional FIB and cryo-FIB are schematically illustrated in Fig. 5. Thinning of a TEM thin foil undergoes a similar process. Cryogenic cooling could also slow down the hydrogen out-diffusion from the specimen during FIB milling, which makes this technique applicable for preparing TEM lamellae and/or APT specimens from hydrogen charged material, and therefore will be an enabler for the investigation of the hydrogen embrittlement behaviour of materials at near-atomic scale[28].

## Discussion

Undesired hydrogen pick-up and hydride formation are usually observed in specimens prepared for microscopic investigations of Ti and its alloys prepared by either electro-polishing, room temperature Ga or Xe Plasma FIB milling. We conducted cryo-PFIB milling to prepare specimens for TEM observation and APT measurements by using a xenon plasma focussed ion beam microscope (FEI Helios) equipped with a cryogenic stage that enabled cooling of the specimen to −135 °C during the final preparation steps. TEM/STEM observation proved that no hydride was formed in CP-Ti after cryo-PFIB preparation. Further APT compositional analysis showed similar results, i.e., no hydride in the CP-Ti and only less than 2 at.% H on average in the β phase of Ti6246 alloy prepared by cryo-PFIB milling. This is almost an order of magnitude less than for specimens prepared by conventional Ga and Xe plasma FIB milling. Through these measurements, we proved cryo-FIB is a promising technique to lower hydrogen pick-up and prevent undesired hydride formation induced during FIB milling in hydride-forming materials such as Ti alloys. Cryo-FIB hence uniquely enables characterization of pure Ti and Ti-based alloys without undesired

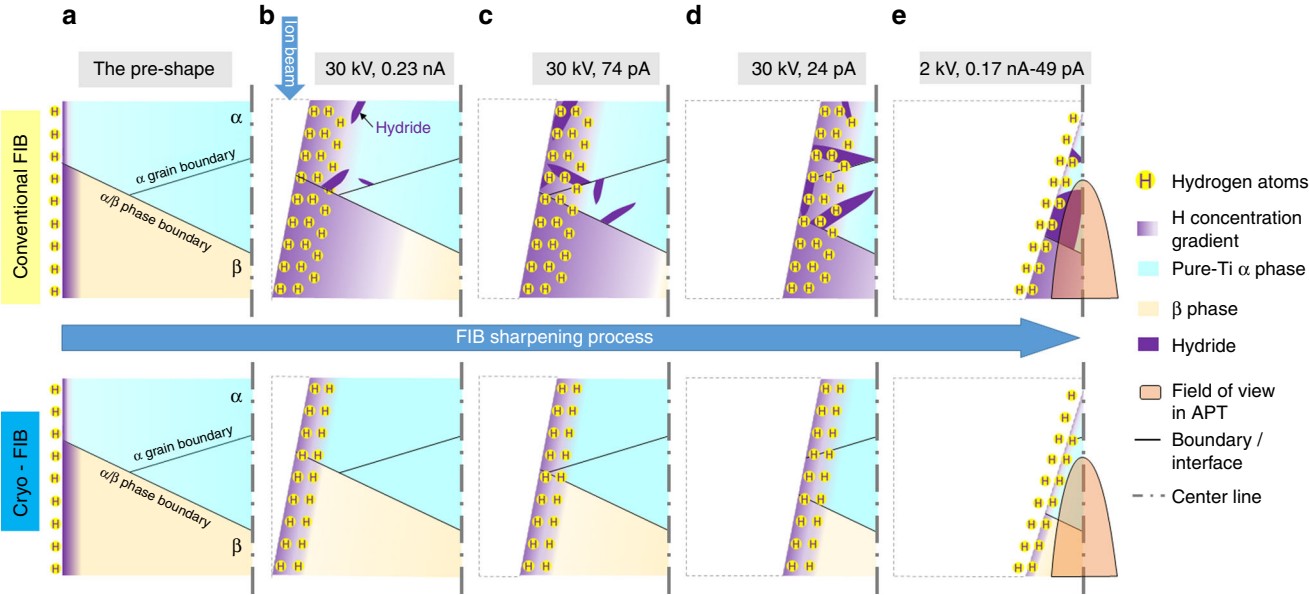

**Fig. 5** Schematic summary: Schematic of H pick-up during focussed ion beam (FIB) sharpening of an atom probe tomography (APT) specimen at room temperature (top) and at cryogenic temperature (bottom). **a** H atoms from environment are adsorbed on the sample surface. **b–d** When applying ion beam milling, H atoms on the surface are implanted by the ion beam. At room temperature, H would then diffuse deep into specimen, and hydrides probably form and grow during milling process in case of pure Ti α phase (hydride formation is likely inhibited by Al addition in α phase[25,35]). At cryogenic temperature, the beam damage and H implantation are slightly reduced and H diffusion is significantly suppressed. **e** Final cleaning at low kV removes the surface damage layer and leaves only a thin layer of H implantation, which is in any case outside the field of view during APT measurement

hydrogen pick-up or hydrides formation during TEM and APT sample preparation. Besides, this technique should also prevent pre-charged hydrogen from diffusing out of a sample and hence enable the investigation of hydrogen embrittlement mechanisms in Ti and its alloys.

If Ti and its alloys are known to be prone to hydrogen pick-up during specimen preparation, which can dramatically affect the results from microstructural characterization, other environmentally sensitive materials are affected by similar issues. Hydrogen and oxygen pick-up by Zr-based alloys and Mg-based alloys for instance are critical for their mechanical properties and their engineering lifetime[52,53]. Hydride formation and phase evolution induced by hydrogen pick-up during specimen preparation has been reported in Zr alloys[11,24,54] and oxygen implantation and oxide layers formed in pure Mg[55]. Preparation artefacts will prevent precise assessment of the microstructure and hence its link to properties for these systems. The protocol we describe herein would alleviate these issues and enable new insights to be gained into some of these long-standing problems.

## Methods

**Cryo-FIB equipment**. A dual-beam scanning electron microscope/focused ion beam (SEM/FIB) FEI Helios Plasma-FIB, making use of Xe ions generated from a plasma source, was fitted with a Dewar and a cold finger (Microscopy Solutions). A solid-state cooling stage was custom designed to accommodate a typical puck that carries APT specimens in a commercial Cameca APT. A large copper bar is mounted inside the SEM to the side of the pole piece of the electron column, to act as a thermal mass; contact between the cryo-stage and the cold finger, and the use of a cryo-shield (not used here), can further help maintain the specimen at low temperature. The cryogenic stage itself is connected to this bar via a stack of copper bands. The cryo-stage is isolated from the main stage of the SEM by a series of vacuum polyether ether ketone (PEEK) spacers. More details on this specific setup can be found in ref. [55].

**Specimen preparation for TEM and APT**. The materials we used were grade 2 commercially pure Ti and a commercial Ti6246 alloy. Both materials were subject to a thermomechanical treatment described in ref. [25]. The bulk samples were mechanically ground using SiC papers to 2500 grit and polished using colloidal silica suspension with 30 vol.% $H_2O_2$ at room temperature.

Twin jet electro-polishing of the 3 mm diameter discs of CP Ti were conducted at −30 °C in a solution of 6% perchloric acid +59% methanol +35% Butoxyethanol. Site-specific lift-out processes for both TEM and APT were conducted at 30 kV and 6–9 nA with Ga and Xe ion sources at ambient temperature. The lift-out TEM lamellae and APT tips were stabilized by Pt deposition (30 KeV, 74 pA ion beam at ambient temperature) on the Cu grid and Si post, respectively. Subsequent thinning and milling were conducted with 30 KeV, 0.46 nA–24 pA Xenon plasma after the stage was cooled to −135 °C. The final milling was done at 2 keV, 24 pA Xe ion beam under cryogenic condition. Conventional Ga and Xe plasma FIB preparation used the same parameters at ambient temperature.

**Specimen handling/transfer**. After the final milling, the room temperature pre-pared specimens were directly transferred from the FIB chamber into the TEM in ambient lab condition within approximately 15 min. The cryo-prepared TEM specimens (at −135 °C) were transferred from the PFIB chamber to a side chamber which is maintained in a high vacuum ($<5 \times 10^{-6}$ Pa) at room temperature so that the specimens can be naturally warmed back to room temperature within 30 min before being transferred to the TEM at ambient temperature and atmospheric pressure. Prior to observation, the entire transfer process takes in the range of 45 min. All specimens were exposed to ambient condition for similar time, approximately 15 min.

The cryo-prepared APT specimens were transferred from the PFIB into the atom probe either at ambient lab conditions, the same procedure as TEM specimens transferred, or under cryogenic ultra-high vacuum conditions using our cryogenic ultra-high vacuum (UHV) sample transfer protocols described in ref. [55]. This process takes approximately 1 h.

**Microstructure characterization and compositional measurement**. The microstructure of TEM specimens prepared by either electro-polishing, room temperature FIB or cryo-PFIB were analysed using an image aberration-corrected FEI Titan Themis 80–300 operated at an accelerating voltage of 300 kV. For phase identification, SAED and HAADF-STEM imaging were conducted. A probe semi-convergence angle of 17 mrad with inner and outer semi-collection angles from 73 to 350 mrad were utilised during HAADF-STEM observation. For LAADF-STEM imaging, a probe semi-convergence angle of 17 mrad and inner and outer semi-collection angles from 14 to 63 mrad were used. Chemical composition at the nanoscale was obtained from APT measurements performed on a Cameca LEAP 5000 XR, operated at a base temperature of 50 K in high-voltage pulsing mode with 20% pulse fraction, 250 kHz pulse frequency and a target detection rate between 5 and 10 ions per 1000 pulses. The pressure in the ultra-high vacuum chamber was consistently below $4 \times 10^{-9}$ Pa.

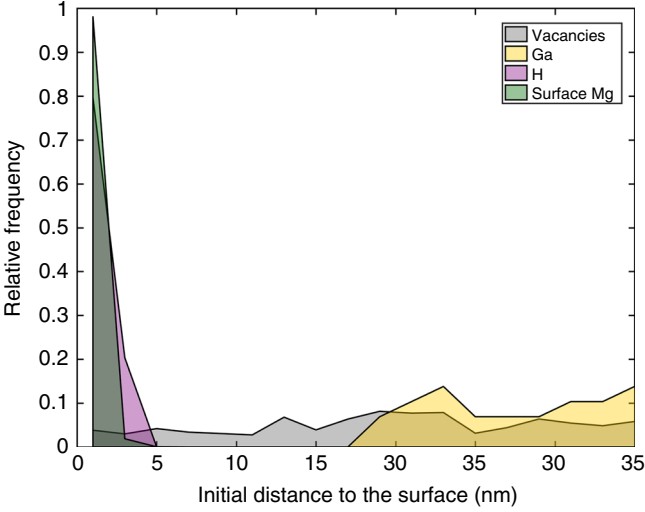

**Fig. 6** Intermixing following implantation in pure Mg. Relative frequency of surface Mg, H, Ga and vacancies as a function of the distance to the surface

**Molecular dynamics simulations**. MD simulations were performed with the open source program LAMMPS (version 11 Aug 2017)[56]. Here we only briefly describe the simulation procedures, but more details are available elsewhere[18,57]. Interactions between Ti atoms were described by an embedded atom method potential, and interactions that include Ga and Xe with the Ziegler–Biersack–Littmark potential[58]. The dimensions $x \times y \times z$ of samples were $12 \times 5 \times 96 \ nm^3$ and $12 \times 5 \times 67 \ nm^3$ for normal and grazing incidence angles, respectively. These were chosen to fit all the collision events and to limit the computational expense. Xe and Ga ions were given a velocity corresponding to the acceleration voltage of 5 kV or 30 kV, from an initial random position above the sample surface, with an angle of 0 or 80 degree to the z-axis. The (0001) surface of the bulk hcp Ti was normal to the z-axis. Periodic boundary conditions (PBCs) were used in the direction x and y, as well as tethered boundary conditions (TBCs) at the 'bottom' in order to prevent any motion of the sample. PBCs are commonly used in MD to mimic an infinite sample. The lengths of periodic directions have however to be chosen carefully to ensure that the interactions between repeating defects are negligible. In the non-periodic direction, the use of TBC allows the system to remains fixed while removing the sharp interface that will come with frozen boundary conditions. Also, ions or primary-knocked atoms (PKAs) reaching the bottom of the sample are able to easily cross TBC and be removed from the simulation as expected, while they will have a high probability to bounce back with frozen boundary condition. Collision cascades were performed within the NVE ensemble (constant energy, constant volume) using a variable time step for at least 15 ps. The default time step is set to 1 fs, and is reduced to ensure that no atom is moving more than 0.005 nm within one time step. A Berendsen thermostat was used to dissipate the heat without interfering directly with the cascade events. An equilibration of 10 ps within the NVT ensemble (constant temperature, constant volume) was performed between each fired ions. The temperature mentioned in our work is calculated based on the average instantaneous atomic velocity. The average velocity is computed by averaging the instantaneous velocity of one atoms with all its neighbours within a distance of 1 nm. Note that this approximation is sufficient for our approach, but a more physically accurate calculation of the atomic temperature should also consider time averaging. MD trajectories were visualized and analysed by using Ovito[59].

**Distribution of species post implantation**. We made the assumption that the Ti redistribution depicts the penetration of H adsorbed on the surface. This implies that, to a first approximation, we assumed that H PKA will behave similarly to Ti PKA under irradiation, especially at low temperature when diffusion is limited. To support this hypothesis, we performed dedicated simulations. Because of a lack of accurate interatomic potential available for modelling the interaction between Ti and H, we used the Mg-H system as surrogate for Ti. A potential for the Mg-H system was recently reported to yield accurate results[60]. Mg and H both adopt a hexagonal-closed packed structure and are hydride formers, admittedly with the formation of Ti hydrides more energetically favourable. This system was irradiated with Ga ions at 30 kV and 138 K with a normal incidence. We then retrieved the distribution of H and Mg atoms within 1 nm of the surface, vacancies (by using a Voronoi/Wigner-Seitz analysis) and implanted Ga ions. All these distributions are normalized to their total count and shown in Fig. 6.

We evidenced here that the profile of H directly follows that of the surface atoms post implantation. The distribution of vacancies does not appear directly correlated with the distribution of neither H nor Ga atoms. The results could be quantitatively different by considering Ti instead of Mg in this simulation, or by changing the temperature to allow for diffusion, but we do not expect the correlation to change significantly, especially not at the temperature at which

experiments were conducted. This set of simulations indicates that the distribution of H atoms follows the distribution of the surface atoms displaced by the irradiation. Therefore, the redistribution of the surface Ti atoms after irradiation is used to mimic the possible implantation of H adsorbed on the surface. In-diffusion of either surface Ti atoms or H atoms subsequent to implantation is not considered here since our MD simulations are constrained within a time range of only several nanoseconds.

**Code availability**. No specific code not widely available was used in this study.

## Data availability

The data that support the findings of this study are available from the corresponding author upon reasonable request.

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

## Acknowledgements

We thank Uwe Tezins and Andreas Sturm for their support to the FIB and APT facilities at MPIE. Y.C. is grateful to the China Scholarship Council (CSC) for funding of her PhD scholarship. Simulations were performed with computing resources granted by RWTH Aachen University under project rwth0297. We are grateful for the financial support from the BMBF via the project UGSLIT and the Max-Planck Gesellschaft via the Laplace project. Erik Bitzek is acknowledged for fruitful discussions. D.D., A.A. and F.D. were funded by Rolls-Royce plc and EPSRC (EP/K034332/1, EP/L025213/1, EP/L015277/1, EP/P029914/1). L.S. and B.G. acknowledge financial support from the ERC-CoG-SHINE-771602.

## Author contributions

Y.C., D.P., D.R. and B.G. designed the study. Y.C. prepared the specimens and performed APT. Y.C.and B.G. interpreted the APT results, with the support of I.M. and X.Z. W.L. performed the TEM and interpreted the results with the help of S.Z. and C.H.L. J.G. and S.K.-K. performed the MD simulations. L.S., A.S., P.K. and I.M. helped with the cryo specimen preparation and the transfer of cryo-transfers. A.A., F.D. and D.D. provided materials and reproduced some experiments at ICL, discussed results and mechanisms. All authors discussed the results, had input and commented on the manuscript.

## Additional information

**Competing interests:** The authors declare no competing interests.

