## [Peer Review File · Nature Communications]

Reviewers' comments:

Reviewer #1 (Remarks to the Author):

The authors compare microscopy observations (transmission electron microscopy [TEM] and atom probe tomography [APT]) obtained after conventional, room temperature specimen preparation methods or cryogenic specimen preparation. The results demonstrate that cryogenic focused ion beam (cryo-FIB) prevents hydrogen ingress and hydride phase precipitation in CP-Ti and Ti6246 alloy. This is an important finding as it identifies a viable specimen preparation pathway for these and other environmentally sensitive samples (e.g. Zr, Mg, or other reactive materials). With this method it will be possible to examine fundamental reaction mechanisms within these materials with greater confidence and quantitative accuracy than has been possible previously. Overall the paper is interesting and well written, but omits critical details for a full validation of the results and reproducibility of its methods. As such significant changes are needed prior to publication. The modeling component of the paper is weak compared to its experimental counterpart. I suggest the MD discussion be reduced in length and instead more attention be given to detailing the experimental methodology and observations.

- Recognition of prior work: It is noted that this is not the first study to employ cryo-FIB to prepare APT specimens (see for example Dumitraschkewitz et al Adv. Eng. Mater. 19 (2016) 1600668, and Schreiber et al Ultramicrosc. 194 (2018) 89-99.), but it is the most thorough comparative study of cryogenic and room temperature specimen preparation artifacts. As such, it will be a useful reference for those who study environmentally sensitive materials with high resolution microscopy techniques. More importantly, it should be noted that the electron microscopy community, particularly in the life sciences, has been using cryo-FIB to prepare specimens for many years. References to these efforts should be added to the introduction.
- There are critical details about the cryogenic preparation methodology and specimen handling that need to be added. The most important of these is timing, particularly for the handling of the cryogenically prepared TEM and APT specimens. The authors make no mention of how the cryo-FIB prepared specimens are extracted from the cryo-FIB to either the TEM or APT. Are the specimens preserved in high vacuum and cryogenic cooling, and if not, how long have the specimens been in ambient lab conditions prior to observation? Do cryo-prepared specimens evolve under ambient conditions outside of the FIB? Have all specimens been exposed to ambient conditions for similar times? These details are essential to identifying the origins of the hydride formation and hydrogen penetration.
- MD Simulations: There are many issues with these simulations and their relevance to the question of hydride formation / hydrogen penetration. In my opinion the paper would be stronger if the MD simulations were removed in their entirety if not significantly improved.
 - o It is unclear why the authors chose only to consider simulations at 30 kV and room temperature when a lower energy (2-5 kV) is intentionally employed experimentally to mitigate the amount of damage at the surface, and changing temperature is apparently THE variable that is leading to a unique experimental result. As such the modeling results neither complement nor supplement the experimental observations and their value to the paper is questionable.
 - o On page 7 the authors claim that the Ti redistribution would “mimic the possible penetration of H-containing species adsorbed on the surface.” It is unclear why the authors expect a self-interstitial distribution of Ti would mimic H, which can readily diffuse interstitially through Ti with or without any radiation induced defects. If a hydrogen atom did absorb the same kinetic energy as Ti, which is equally unlikely given its small nuclear size, it would also channel much deeper into the material than did Ti. As such the presented calculations seem irrelevant to the discussion of the hydrogen distribution. Radiation-induced vacancies are also potential trapping sites for H, which is arguably more important for H incorporation, but are not discussed.

o Why do the authors believe that the Ti redistribution is the most relevant marker for “FIB-induced structural damage” and how does it relate to a H distribution? I could perhaps see this damaged region being more reactive to forming hydrides, but I see no direct correlation to either H solubility or diffusivity. What is the basis for this assumed correlation? I would assume the Xe or Ga implantation profiles would be just as relevant as the Ti redistribution, while vacancy distributions are probably more important still.

o On page 9 the authors conclude that the beam damage is insensitive to sample temperature. This statement seems to be based on the similar level of extreme localized heating during irradiation that is quickly quenched back to room temperature. This conclusion neglects the behavior of the radiation-induced defects after the initial damage cascade. In particular, the ability of defects (vacancies and interstitials) to diffuse and annihilate at the surface of the nanoscopic APT needle. These processes are inherently temperature dependent and should be directly addressed and not brushed aside. More importantly, the theme of the paper is cryo-FIB matters. If that is the case, why are temperature effects dismissed so quickly and without validation?

- The discussion of H ingress in section 3.4, and the corresponding Fig 5, assume that specimens prepared at room temperature are structurally identical to those produced cryogenically and that the only relevant variable is diffusivity differences. A missing component in this discussion is whether or not cryo-prepared specimens, subsequently left in ambient conditions, precipitate hydrides. Or put another way, does hydride precipitation require concurrent irradiation and room temperature, or only room temperature and time? This seems to be a fundamental and critical question for developing a reliable protocol for producing and handling these delicate specimens. If the only relevant difference is diffusivity, should not the cryo-prepared specimens evolve at ambient conditions after preparation? Or if it is in fact the combination of irradiation and temperature, that needs to be proven.

Minor Comments:

- There is a systematic difference in the size of the collected APT datasets between cryogenic and room temperature-prepared specimens. This is problematic when the authors are trying to prove hydrides do not form when looking at a substantially smaller volume of material. The authors should mention this difference and indicate how they are confident in their interpretation even with a reduced volume of material analyzed.
- The authors alternate between Xe-PFIB, PFIB and Plasma FIB throughout the manuscript. For simplicity I suggest utilizing one acronym.

Reviewer #2 (Remarks to the Author):

In this paper, cryo-FIB milling has been proposed by the authors to control the undesired hydrogen pick-up and hydride formation in CP Ti and Ti6246 alloys during TEM and APT sample preparations. Both simulation and experiments have been carried out to demonstrate the feasibility and effectiveness of the proposed method. The manuscript is generally in good structure and the authors have discussed the results including the potential source of Hydrogen, FIB-induced structural damage, beam induced heating effects, and the H ingress.

Based on its current presentation, minor revision is recommended before the acceptance.

- (1) the abstract might need to mention the MD simulation;
- (2) the introduction can be further shortened;
- (3) Please provide more details on MD simulation parameters and how those parameters were selected. For example, the boundary conditions and the potential effects of the using period boundary condition in MD simulation? What's the timestep specified in the simulation and what variable timestep were used? How the T has been calculated in the MD simulation?

(4) the caption of Fig.3(e) is a little bit confused, 'Position of the Ti atoms after irradiation as a function of their position before irradiation'?

(5) In section 6.4, the author has introduced the MD simulation and the T was kept 300K using NVT ensemble. However, the author mentioned in Page 11 line 309-312 that, 'Our MD simulation indicates that the base temperature makes no difference to beam damage and possible H implantation on the surface layer, but by suppressing long-range diffusion of H, cryogenic cooling prevents hydrogen accumulation in β -Ti and hydride formation in α -Ti during FIB milling.' Please provide more results to support this argument. It is not clear whether the ion bombardment at -135 °C have been tried in the MD simulation or not? If so, how the T was controlled in the simulation?

Editorial Note: Reviewer #3 included two papers as attachments for reference. The full articles have not been included in this Peer Review file, but the complete references have been inserted.

Reviewer #3 (Remarks to the Author):

Review comments: Minor Revisions

Title: Cryogenic focused ion beam milling of environmentally-sensitive materials: example of Ti and its alloys

Article Type: Research Paper

I appreciate the progress and effort that the authors have made. I suggest this manuscript to be minor revised before acceptance for publication.

1. According to Colas et al. [1] and Steuwer et al. [2], the hydrogenation behavior of Zr and its alloys would be strongly correlated to the local stresses. As reported in Ref.24 "the stress relaxation accompanied by the lamella deformation generated during the conventional FIB milling processing provides the driving force for the Zr hydrogenation, when take appropriate thinning procedure no clear deformation and no platelets were observed in the lamella."

So, does the relaxation of inner stress influence the hydrogen diffusivity in Ti alloys?

[1]: K.B. Colas, A.T. Motta, J.D. Almer, M.R. Daymond, M. Kerr, A.D. Banchik, P. Vizcaino, J.R. Santisteban, 'In situ study of hydride precipitation kinetics and re-orientation in Zircaloy using synchrotron radiation', *Acta Materialia*, Volume 58, Issue 20, 2010, Pages 6575-6583, ISSN 1359-6454,

<https://doi.org/10.1016/j.actamat.2010.07.018>.

(<http://www.sciencedirect.com/science/article/pii/S1359645410004520>)

[2]: A. Steuwer, J.R. Santisteban, M. Preuss, M.J. Peel, T. Buslaps, M. Harada, 'Evidence of stress-induced hydrogen ordering in zirconium hydrides', *Acta Materialia*, Volume 57, Issue 1, 2009, Pages 145-152, ISSN 1359-6454,

<https://doi.org/10.1016/j.actamat.2008.08.061>.

(<http://www.sciencedirect.com/science/article/pii/S1359645408006332>)

Reviewers' comments:

Reviewer #1 (Remarks to the Author):

The authors compare microscopy observations (transmission electron microscopy [TEM] and atom probe tomography [APT]) obtained after conventional, room temperature specimen preparation methods or cryogenic specimen preparation. The results demonstrate that cryogenic focused ion beam (cryo-FIB) prevents hydrogen ingress and hydride phase precipitation in CP-Ti and Ti6246 alloy. This is an important finding as it identifies a viable specimen preparation pathway for these and other environmentally sensitive samples (e.g. Zr, Mg, or other reactive materials). With this method it will be possible to examine fundamental reaction mechanisms within these materials with greater confidence and quantitative accuracy than has been possible previously. Overall the paper is interesting and well written, but omits critical details for a full validation of the results and reproducibility of its methods. As such significant changes are needed prior to publication. The modeling component of the paper is weak compared to its experimental counterpart. I suggest the MD discussion be reduced in length and instead more attention be given to detailing the experimental methodology and observations.

- Recognition of prior work: It is noted that this is not the first study to employ cryo-FIB to prepare APT specimens (see for example Dumitraschkewitz et al *Adv. Eng. Mater.* 19 (2016) 1600668, and Schreiber et al *Ultramicrosc.* 194 (2018) 89-99.), but it is the most thorough comparative study of cryogenic and room temperature specimen preparation artifacts. As such, it will be a useful reference for those who study environmentally sensitive materials with high resolution microscopy techniques. More importantly, it should be noted that the electron microscopy community, particularly in the life sciences, has been using cryo-FIB to prepare specimens for many years. References to these efforts should be added to the introduction.

We thank the reviewer for pointing this shortcoming. We have added relevant references, e.g.:

- Marko, M., Hsieh, C., Schalek, R., Frank, J. & Mannella, C. Focused-ion-beam thinning of frozen-hydrated biological specimens for cryo-electron microscopy. *Nat. Methods* 4, 215–217 (2007).
- Schreiber, D. K., Perea, D. E., Ryan, J. V., Evans, J. E. & Vienna, J. D. A method for site-specific and cryogenic specimen fabrication of liquid/solid interfaces for atom probe tomography. *Ultramicroscopy* 194, 89–99 (2018).
- Dumitraschkewitz, P., Gerstl, S. S. A., Uggowitz, P. J., Löffler, J. F. & Pogatscher, S. Atom Probe Tomography Study of As-Quenched Al–Mg–Si Alloys. *Adv. Eng. Mater.* 19, 1–5 (2017).

This sentence was added to the introduction:

"Cryo-FIB milling and specimen handling has been extensively used in the biological sciences for electron microscopy investigations³⁰, and is increasingly applied for atom probe investigations^{31,32}."

- There are critical details about the cryogenic preparation methodology and specimen handling that need to be added. The most important of these is timing, particularly for the handling of the cryogenically prepared TEM and APT specimens. The authors make no mention of how the cryo-FIB prepared specimens are extracted from the cryo-FIB to either the TEM or APT. Are the specimens preserved in high vacuum and cryogenic cooling, and if not, how long have the specimens been in ambient lab conditions prior to observation? Do cryo-prepared specimens evolve under ambient conditions outside of the FIB? Have all specimens been exposed to ambient conditions for similar times? These details are essential to identifying the origins of the hydride formation and hydrogen penetration.

We sincerely thank the reviewer for pointing out this missing part in the manuscript.

The experimental conditions for cryo-FIB preparation have been added in the second paragraph in section 6.2:

Cryogenic preparation:

Site-specific lift-out processes for both TEM and APT were conducted at 30kV and 6–9nA with Ga and Xe ion sources at ambient temperature. Subsequent thinning and milling were conducted with 30KeV, 0.46 nA~24 pA Xenon plasma after the stage was cooled to -135°C. The final milling was done at 2 keV, 24 pA Xe ion beam under cryogenic condition.

Details of the specimen handling/transfer were also added to a newly created Methods Section 6.3 in the manuscript:

Specimen handling/transfer:

After final milling, the room-temperature prepared specimens were directly transferred from the FIB chamber into the TEM in ambient lab condition within approx. 15 min. The cryo-prepared TEM specimens (at -135°C) were transferred from the PFIB chamber to a side-chamber which is maintained in a high vacuum ($< 5 \times 10^{-6}$ Pa) at room temperature so that the specimens can be naturally warmed back to room temperature within 30 min before being transferred to the TEM at ambient temperature and atmospheric pressure. Prior to observation, the entire transfer process takes in the range of 45 min. All specimens were exposed to ambient condition for similar time, approx. 15min.

The cryo-prepared APT specimens were transferred from the PFIB into the atom probe either at ambient lab conditions, the same procedure as TEM specimens transferred, or under cryogenic ultra-high vacuum conditions using our cryogenic ultra-high vacuum (UHV) sample transfer protocols described in ref. 56. This process takes approx. 1 hour.

The measured amount of H in the α and/or β phase of CP-Ti and Ti6246 alloy specimens, **transferred via either route** is consistently below 2 at.%, and no hydride were detected, as listed in Table 1. When exposed to ambient conditions over 24 hours, a thin oxygen-rich layer, approx. 10nm, on the surface can be detected for APT specimens, the same problem as pure Mg specimens, as addressed in ref. 56.

Overall, these observations supports our statement that significant hydrogen pick-up happens during conventional FIB milling at room temperature and not during transfer.

- MD Simulations: There are many issues with these simulations and their relevance to the question of hydride formation / hydrogen penetration. In my opinion the paper would be stronger if the MD simulations were removed in their entirety if not significantly improved.

o It is unclear why the authors chose only to consider simulations at 30 kV and room temperature when a lower energy (2-5 kV) is intentionally employed experimentally to mitigate the amount of damage at the surface, and changing temperature is apparently THE variable that is leading to a unique experimental result. As such the modeling results neither complement nor supplement the experimental observations and their value to the paper is questionable.

We thank the reviewer for the accurate comment on these two points. We have performed additional simulations regarding them. The results are shown as follows.

Simulation on low energy irradiation. we performed simulation for Ga irradiation with an acceleration voltage of 5 kV. As shown in Fig. 1, reducing the amount of deposited energy could lead to a reduction of the observed re-distribution of surface atoms after implantation.

However, as mentioned in the manuscript, MD simulations are constrained to short time scales, with a consequence of a dramatically small doses by comparison to experiment. Therefore, in order to increase the total energy deposited on the surface by the ionic irradiation, we decided to use the higher acceleration voltage, which is still typical of the values used for preparation. In the following simulations investigating the effect of the base temperature of the substrate, we maintained the acceleration voltage at 30kV to enhance the magnitude of the investigated phenomena.

Simulations at cryogenic temperature. New simulations were performed for both Xe^+ and Ga^+ irradiation at 138K (-135°C). The results, shown in the figure below, indicate that the implantation of surface atom is slightly reduced for both Xe and Ga irradiation when decreasing the base temperature of the substrate.

The above two new figures have been added to Fig.3, and the following text is added to the Results Section 2.3 in the manuscript.

Figure 3(c) shows a similar graph for simulations performed for singly charged Ga ions accelerated to 5kV and 30 kV, both at normal incidence. With a lower incoming energy, the spread in the distribution of the surface atoms is reduced, as expected from previously published studies^{40,41}. Only a low dose can be deposited during the short time of a simulation. It could hence be argued that the total energy deposited into the surface by the ionic irradiation in the MD simulations is too low to be representative. We hence decided to only use the higher acceleration voltage (30kV) for our other simulations. Simulations investigating the effect of the base temperature of the substrate were performed with an acceleration voltage of 30kV, the results shown in Figure 3(d). Decreasing the base temperature of the substrate leads to a slight reduction of the observed redistribution of the surface atoms for both Xe and Ga ions. Previous

results from simulations of FIB milling reported extremely high peak temperatures localised where the incoming ions penetrate the metal 42, and our simulations exhibit similar local peak temperatures largely above the melting point but quenched back to room temperature within approximately 2ps, and even faster at low temperature, resulting in a reduced level of damage.

o On page 7 the authors claim that the Ti redistribution would “mimic the possible penetration of H-containing species adsorbed on the surface.” It is unclear why the authors expect a self-interstitial distribution of Ti would mimic H, which can readily diffuse interstitially through Ti with or without any radiation induced defects. If a hydrogen atom did absorb the same kinetic energy as Ti, which is equally unlikely given its small nuclear size, it would also channel much deeper into the material than did Ti. As such the presented calculations seem irrelevant to the discussion of the hydrogen distribution. Radiation-induced vacancies are also potential trapping sites for H, which is arguably more important for H incorporation, but are not discussed.

The reviewer #1 raised here an interesting, and even critical, point.

We made the assumption that the Ti redistribution depicts the penetration of H adsorbed on the surface. This implies that, to a first approximation, we assumed that H PKA will behave similarly to Ti-PKA under irradiation, especially at low temperature when diffusion is limited. To support this hypothesis, we performed dedicated simulations.

Because of a lack of accurate interatomic potential available for modelling the interaction between Ti and H, we used the Mg-H system as surrogate for Ti. Mg and Ti both adopt a hexagonal-closed packed structure and are hydride formers, admittedly with the formation of Ti-hydrides more energetically favourable. Fig. 3 illustrates the initial configuration for the simulations where Mg replaces Ti, and we introduced interstitial H within 1 nm of the surface. (Perspective and side views. Mg in green and H in red. System equilibrated at 138K.):

This system is irradiated with Ga ions at 30kV with a normal incidence. We then retrieved the distribution of:

- H atoms.
- Mg atoms that were located within 1 nm to the surface, ie. The surface atoms.
- The vacancies (by using a Voronoi/Wigner-Seitz analysis).
- The implanted Ga ions.

All these distributions are normalized to their total count and shown in the Fig. 4:

As evidenced here, the profile of H following implantation is similar to the implantation profile of the surface atoms. The distribution of vacancies appears not correlated with the distribution of neither H nor Ga atoms. The results could be quantitatively different by considering Ti instead of Mg in this simulation, or by changing the temperature to allow for diffusion, but we do not expect the correlation to change significantly, especially not at the temperature at which experiments were conducted. This simulation strongly support our initial hypothesis that the distribution of H atoms follows the distribution of the surface atoms displaced by the irradiation. Therefore, the redistribution of the surface Ti atoms after irradiation is used to mimic the possible implantation of H adsorbed on the surface. Note that in-diffusion of either surface Ti atoms or H atoms subsequent to implantation is not considered here since our MD simulations are constrained within a time range of only several nanoseconds. The approximation we propose in our manuscript appears correct within limitation of MD simulations.

The above figure and text were introduced as a **new section 6.6 “Distribution of species post-implantation”** to the “Methods” section in the manuscript.

o Why do the authors believe that the Ti redistribution is the most relevant marker for “FIB-induced structural damage” and how does it relate to a H distribution? I could perhaps see this damaged region being more reactive to forming hydrides, but I see no direct correlation to either H solubility or diffusivity. What is the basis for this assumed correlation? I would assume the Xe or Ga implantation profiles would be just as relevant as the Ti redistribution, while vacancy distributions are probably more important still. **We thank the Reviewer #1 to allow us to precise our results. The figure above actually addresses the points raised. It appears clearly that the distribution of H is related to the distribution of surface atoms, and not to the distribution of vacancies or implanted Ga ions.**

o On page 9 the authors conclude that the beam damage is insensitive to sample temperature. This statement seems to be based on the similar level of extreme localized heating during irradiation that is quickly quenched back to room temperature. This conclusion neglects the behavior of the radiation-induced defects after the initial damage cascade. In particular, the ability of defects (vacancies and interstitials) to diffuse and annihilate at the surface of the nanoscopic APT needle. These processes are inherently temperature dependent and should be directly addressed and not brushed aside. More importantly, the theme of the paper is cryo-FIB matters. If that is the case, why are temperature effects dismissed so quickly and without validation?

We really appreciate this critical comment by the Reviewer. It is true that we neglected the effect that vacancies may have on the ingress of hydrogen. The reviewer is right to point to the possible, and even likely annihilation of radiation defects on the surface of the APT specimen, which has a large surface to

volume ratio. In-diffusion of radiation-induced defects, especially vacancies which are potential trapping site for H, would assist the introduction of H below the sputtered surface. However, since H diffuses via an interstitial mechanism, the influence of these crystalline defects on the H long-range diffusion is expected to be insignificant at room temperature, although might be slightly enhanced by beam-induced heating. Therefore, the fast diffusion of H is reckoned to play the critical role in the H ingress during conventional FIB milling.

At cryogenic temperature, beam damage and possible H implantation is confined within 5 nm on the surface, indicated by the new MD simulation results presented above, and the mobility of the radiation-induced defects is expected to be constrained as well. Most importantly, by suppressing H diffusivity, cryo-cooling substantially inhibits undesired in-diffusion of hydrogen from the environment in both α - and β -Ti, and hence prevent hydride formation in α -Ti and hydrogen accumulation in β -Ti during FIB milling.

These aspects were blended into the discussion within the article and highlighted in red and Fig.5 in the manuscript were modified taking into account the new MD simulation results that the beam implantation is slightly reduced at cryogenic temperature.

- The discussion of H ingress in section 3.4, and the corresponding Fig 5, assume that specimens prepared at room temperature are structurally identical to those produced cryogenically and that the only relevant variable is diffusivity differences. A missing component in this discussion is whether or not cryo-prepared specimens, subsequently left in ambient conditions, precipitate hydrides. Or put another way, does hydride precipitation require concurrent irradiation and room temperature, or only room temperature and time? This seems to be a fundamental and critical question for developing a reliable protocol for producing and handling these delicate specimens. If the only relevant difference is diffusivity, should not the cryo-prepared specimens evolve at ambient conditions after preparation? Or if it is in fact the combination of irradiation and temperature, that needs to be proven.

We thank the Reviewer for pointing out this missing component which stems from a lack of clarity on our side.

No hydride precipitates and constantly less than 2 at.% H are detected in both the α phase and β phase of CP-Ti and Ti6246 specimens prepared by cryo-PFIB, transferred either at ambient lab condition or under cryogenic ultra-high vacuum conditions. (This has been clarified in the Result Section 2.2)

Two essential aspects of H ingress during conventional FIB milling. First, the sources of H can be the decomposition of hydrocarbon, moisture and organometallic precursor stimulated by the energetic electron/ion beam during milling. Second, the H introduction assisted by the beam implantation and the high density of radiation-induced defects, and more importantly, the fast in-diffusion at room temperature. The specimens prepared under cryo-conditions and left under ambient conditions of temperature and pressure do not show hydrides due to the limited intake of H, as discussed in the article. However, specimens easily develop a surface oxide. Similarly, as demonstrated for pure Mg samples in ref. 56, a thin oxygen-rich layer, approx. 10nm, on the specimen's surface can be detected for APT specimens when exposed to ambient conditions over 24 hours.

Minor Comments:

- There is a systematic difference in the size of the collected APT datasets between cryogenic and room temperature-prepared specimens. This is problematic when the authors are trying to prove hydrides do not form when looking at a substantially smaller volume of material. The authors should mention this

difference and indicate how they are confident in their interpretation even with a reduced volume of material analyzed.

The Reviewer #1 raised a critical point here. The size of the collected APT datasets from cryo-prepared specimens are significantly smaller than those from conventional prepared ones, particularly for CP-Ti. A reason is that, experimentally, α -Ti often has a much lower yield than the hydride phase (only present in room temperature-prepared specimens) during APT measurements with high-voltage mode which we intentionally applied for all measurements in this work to avoid the artificial variation of H composition induced by the interaction between laser and specimen^{1,2}. The presence of H is likely to reduce the electrostatic field required to cause field evaporation of the surface atoms – this was investigated by E.W. Müller in the early days of the technique by introducing gas in the chamber (see J Appl. Phys. 36(8) 2497).

However, we are quite confident in our interpretation based on other two observations: first and foremost no hydride phase has ever been observed in cryo-prepared CP-Ti TEM lamellae, which has a much larger field of view – this was one of the main motivations to perform TEM, as shown in Fig. 1d; second, significantly lower H composition, i.e. comparable to the background level, is detected in the β phase in all cryo-prepared Ti6246 APT specimens, as shown in Fig. 2d, which exhibit much larger analysed volumes.

- The authors alternate between Xe-PFIB, PFIB and Plasma FIB throughout the manuscript. For simplicity I suggest utilizing one acronym.

We appreciate the comment by the Reviewer.

Reviewer #2 (Remarks to the Author):

In this paper, cryo-FIB milling has been proposed by the authors to control the undesired hydrogen pick-up and hydride formation in CP Ti and Ti6246 alloys during TEM and APT sample preparations. Both simulation and experiments have been carried out to demonstrate the feasibility and effectiveness of the proposed method. The manuscript is generally in good structure and the authors have discussed the results including the potential source of Hydrogen, FIB-induced structural damage, beam induced heating effects, and the H ingress.

Based on its current presentation, minor revision is recommended before the acceptance.

(1) the abstract might need to mention the MD simulation;

We appreciate the comment by the Reviewer. The end of the abstract now reads:

The underlying mechanisms for H ingress are discussed, and supported by molecular dynamics (MD). We show that by significantly lowering the thermal activation for diffusion of H, the technique inhibits undesired hydrogen pick-up from the environment during preparation and prevents the pre-charged hydrogen from diffusing out of the sample, and hence allows us to investigate the hydrogen embrittlement mechanisms of Ti based alloys at the nanoscale.

(2) the introduction can be further shortened;

(3) Please provide more details on MD simulation parameters and how those parameters were selected. For example, the **boundary conditions** and the **potential effects of the using period boundary condition** in MD simulation? What's the **timestep** specified in the simulation and what **variable timestep** were used? How the **T has been calculated in the MD simulation**?

We thank Reviewer #2 for allowing us clarified our methods. We extended the method section.

(4) the caption of Fig.3(e) is a little bit confused, 'Position of the Ti atoms after irradiation as a function of their position before irradiation'?

We thank the Reviewer for pointing this out. We adapted the caption to hopefully clarified it: *“Ti atoms: position after irradiation (final distance to surface) as a function position before irradiation (initial distance to surface).”*

(5) In section 6.4, the author has introduced the MD simulation and the T was kept 300K using NVT ensemble. However, the author mentioned in Page 11 line 309-312 that, 'Our MD simulation indicates that the base temperature makes no difference to beam damage and possible H implantation on the surface layer, but by suppressing long-range diffusion of H, cryogenic cooling prevents hydrogen accumulation in β -Ti and hydride formation in α -Ti during FIB milling.' Please provide more results to support this argument. It is not clear whether the ion bombardment at -135 °C have been tried in the MD simulation or not? If so, how the T was controlled in the simulation?

We sincerely thank the Reviewer to allow us to provide more results. We have performed additional simulations for both Xe and Ga ionic irradiation at -135°C, the results shown in the following figure. These new simulations indicate that the implantation of surface atom is slightly reduced for both Xe and Ga irradiation when decreasing the base temperature of the substrate.

These aspects were blended into the discussion within the article and highlighted in red. Fig.5 in the manuscript was modified to account for the new MD simulation results. We extended the method section to provide more details about MD simulation parameters and how these parameters are selected.

Reviewer #3 (Remarks to the Author):

Review comments: Minor Revisions

Title: Cryogenic focused ion beam milling of environmentally-sensitive materials: example of Ti and its alloys

Article Type: Research Paper

I appreciate the progress and effort that the authors have made. I suggest this manuscript to be minor revised before acceptance for publication.

1. According to Colas et al. [1] and Steuwer et al. [2], the hydrogenation behavior of Zr and its alloys would be strongly correlated to the local stresses. As reported in Ref.24 “the stress relaxation accompanied by the lamella deformation generated during the conventional FIB milling processing provides the driving force for the Zr hydrogenation, when take appropriate thinning procedure no clear deformation and no platelets were observed in the lamella.”

So, does the relaxation of inner stress influence the hydrogen diffusivity in Ti alloys?

We appreciate the Reviewer’s comment and sharing the references. Indeed, the stress state does strongly affect the hydrogenation behavior, in particular the hydride distribution and orientation, in Zr alloys according to Colas et al. and Steuwer et al. In ref.24, large number of hydride platelets, formed both at the grain boundaries and in the grain interior in the lamella during ion milling, are preferentially orientated. Hence, it is very likely that the inner stress state played an important role in the hydrogenation behavior during conventional FIB milling. This phenomenon, however, is not commonly observed in our work on CP-Ti and Ti alloys. In most cases, hydrides, formed during conventional FIB milling, are observed at α/β phase boundaries or α grain boundaries that can act as easy diffusion paths for H. For example, the HAADF image in Fig. 1c shows two hydride platelets nucleated at the grain

boundary and grown into the interior of the two adjacent grains in CP-Ti. In binary Ti-2wt.% Fe and Ti-4 wt.% Mo alloys, hydrides are often observed at the low-angle grain boundaries of α phase and α/β phase boundaries, as reported in our previous work, ref.25. No preferential orientation was observed for the hydrides formed during conventional FIB milling, either at grain boundaries or in the grain interior in CP-Ti and binary Ti alloys. Therefore, there is no direct evidence to suggest that the relaxation of inner stress influences the hydrogen diffusivity in Ti alloys. Of course, it should be acknowledged that appropriate thinning procedure may avoid hydride precipitation in both Zr and Ti alloys, as referred by the Reviewer. But our work does suggest that hydride formation can be, most easily and entirely, suppressed by cryo-PFIB milling.

References

1. Sundell, G., Thuvander, M. & Andrén, H. O. Hydrogen analysis in APT: Methods to control adsorption and dissociation of H₂. *Ultramicroscopy* **132**, 285–289 (2013).
2. Kolli, R. P. Controlling residual hydrogen gas in mass spectra during pulsed laser atom probe tomography. *Adv. Struct. Chem. Imaging* **3**, 10 (2017).

REVIEWERS' COMMENTS:

Reviewer #1 (Remarks to the Author):

I would like to thank the authors for providing robust answers to my questions.

In my opinion, the authors' explanation of the origins of the hydride formation during room temperature specimen preparation is still incomplete. I cannot believe that diffusion differences from room temperature to -135C are sufficient to explain the behavior, but the authors have made their case in a systematic and repeatable manner that can be tested and expanded upon by others in future studies. The APT community will take notice of this work and it will prompt further discussion and progress on best practices for specimen preparation and the interpretation of hydrogen-related artifacts in other materials systems.

- Daniel Schreiber

Reviewer #2 (Remarks to the Author):

As far as I concerned, the paper is interesting and well written. The critical details missed have been added in the revised version and all review comments have been addressed. I am happy with the revision and would suggest the acceptance of the manuscript.

Reviewer #3 (Remarks to the Author):

This manuscript had been revised carefully and the quality meets the requirements, I suggest the article can be published without further modification.